# The Role of White Matter Disconnection in the Symptoms Relating to the Anarchic Hand Syndrome: A Single Case Study

**DOI:** 10.3390/brainsci11050632

**Published:** 2021-05-14

**Authors:** Valentina Pacella, Giuseppe Kenneth Ricciardi, Silvia Bonadiman, Elisabetta Verzini, Federica Faraoni, Michele Scandola, Valentina Moro

**Affiliations:** 1Npsy.Lab-VR, Dipartimento di Scienze Umane, Università di Verona, Lungadige Porta Vittoria, 17-37129 Verona, Italy; valentina.pacella.90@gmail.com (V.P.); federica.faraoni3@gmail.com (F.F.); michele.scandola@univr.it (M.S.); 2Brain Connectivity and Behaviour Laboratory, Sorbonne Universities, 75006 Paris, France; 3Groupe d’Imagerie Neurofonctionnelle, Institut des Maladies Neurodégénératives-UMR 5293, CNRS, CEA University of Bordeaux, 33076 Bordeaux, France; 4Neuroradiology, AOUVR Azienda Ospedaliera Universitaria Integrata di Verona, Piazzale Stefani, 1-37126 Verona, Italy; giuseppe.ricciardi@aovr.veneto.it; 5IRCSS, Ospedale Sacro Cuore-Don Calabria, Via Rizzardi, 4-37024 Negrar di Valpolicella, Italy; silvia.bonadiman@sacrocuore.it (S.B.); elisabetta.verzini@sacrocuore.it (E.V.)

**Keywords:** anarchic hand syndrome, DTI, white matter disconnection, lesion mapping, sense of agency, posterior lesions

## Abstract

The anarchic hand syndrome refers to an inability to control the movements of one’s own hand, which acts as if it has a will of its own. The symptoms may differ depending on whether the brain lesion is anterior, posterior, callosal or subcortical, but the relative classifications are not conclusive. This study investigates the role of white matter disconnections in a patient whose symptoms are inconsistent with the mapping of the lesion site. A repeated neuropsychological investigation was associated with a review of the literature on the topic to identify the frequency of various different symptoms relating to this syndrome. Furthermore, an analysis of the neuroimaging regarding structural connectivity allowed us to investigate the grey matter lesions and white matter disconnections. The results indicated that some of the patient’s symptoms were associated with structures that, although not directly damaged, were dysfunctional due to a disconnection in their networks. This suggests that the anarchic hand may be considered as a disconnection syndrome involving the integration of multiple antero-posterior, insular and interhemispheric networks. In order to comprehend this rare syndrome better, the clinical and neuroimaging data need to be integrated with the clinical reports available in the literature on this topic.

## 1. Introduction

The anarchic hand syndrome (AHS) is a rare neurological condition that is characterised by smooth, involuntary, goal-directed movements of an upper limb, which are executed without volitional control [1]. Typical clinical manifestations involve hand grasping or groping, the compulsive manipulation of tools, an inability to enact motor command, inter-manual conflict and arm levitation [2]. Patients sometimes report a sense of alienness with respect to their arm, but in general they recognise that the affected hand is part of their own body. In fact, although the two terms are sometimes used as synonyms, a distinction has been shown between AHS and alien hand syndrome [3,4]. 

AHS may arise after the occurrence of a lesion in either the right or left hemisphere of the brain that usually encompasses the corpus callosum and impairs interhemispheric connections. Various cortical and subcortical lesions (in frontal or parietal areas, and the corpus callosum or thalamus, see [2] for a review) involve a variety of different combinations of symptoms, thereby increasing the complexity of the clinical diagnosis and theoretical explanations for AHS.

Frontal damage usually involves the supplementary motor area, the premotor cortex and the anterior cingulate cortex and results in apparently purposeful movements and environment-dependent behaviours, such as grasping, groping, compulsive manipulation of tools, unresponsiveness and magnetic apraxia [2,5,6]. According to the dual premotor system theory [7], these frontal symptoms are caused by an unbalance between the two mechanisms relating to action control—the medial premotor system and the premotor lateral cortex (PMC). The medial premotor system encompasses the supplementary motor area (SMA) and the anterior cingulate cortex (ACC) and is responsible for internally driven activity. This also exerts an inhibitory activity over the PMC [8] which is instead involved in environmentally or externally driven actions. The impaired inhibitory activity of the SMA over the PMC triggers, or at least releases, uncontrolled, environmentally driven behaviours and automatic responses to external stimuli [7].

As SMA is involved in the process of action selection, AHS has also been considered as a disorder of the intentions to act [9]. Recently, this hypothesis has been further explored by Pacella and Moro [10] who suggest that AHS may be considered to be a deficit in the awareness of motor intention.

AHS has also been reported in patients suffering from parieto-occipital lesions [11,12,13]. In these cases, the syndrome is typically associated with spontaneous, non-purposeful movements such as levitation and repetitive movements, sensory loss, self-grabbing and feelings denoting the personification of the hand [2]. Furthermore, parietal lesions have been associated with abnormal judgments of the sense of agency [12]. However, non-purposeful movements, along with sensory and dexterity loss, have also been described after lesions to subcortical structures such as the basal ganglia and thalamus.

Typically, the disconnections relating to symptoms of AHS are due to callosal lesions, which lead to inter-manual conflict, diagnostic dyspraxia, unilateral apraxia, a lack of bimanual coordination, loss of dexterity and rhythm disorders. These symptoms have been associated with a failure to inhibit the non-dominant hemisphere during tasks that require dominant-hemisphere motor control or verbal mediation [5].

Despite the classification of AHS as being associated with the anterior, posterior or callosal regions, a review of the literature on the topic [2,14] indicates that there is no clear distinction in the manifestations of AHS based on the lesion site, since the same symptoms sometimes derive from apparently different lesions (i.e., patients with posterior damage may present with typically frontal symptoms and vice versa). There are two possible reasons for this inconsistency.

Firstly, AHS is a rare syndrome that patients often recover from in the first few days after the lesion onset. This means that any descriptions of the symptoms may be incomplete and limited to the signs that spontaneously emerge in the patient’s behaviour, and there is no systematic investigation of any less evident symptoms. Secondly, until now, the anatomical correlates of AHS have been analysed by means of the traditional lesion-symptom mapping approach, a procedure that focuses on direct lesions but precludes the investigation of brain disconnections other than the callosal damage.

In the present study, our aim was to overcome these limitations and take advantage of recent methodological advances in the neuroimaging of structural connectivity that have already proven to be fruitful in the study of neuropsychological deficits [15,16,17]. The objective was to investigate the contribution of white matter tract disconnections in the clinical manifestation of AHS. An AHS patient with damage to the posterior right hemisphere was submitted to an in-depth, repeated investigation of her symptoms. The assessment was based on a classification of symptoms resulting from a review of the literature [2] and an update of the review which includes more recent studies. Four clinical assessments were carried out in order to record the evolution of the patient’s symptoms over the six-month period following the lesion onset. In addition, an anatomical study was conducted which allowed us to integrate the data relating to the direct lesions and the white matter disconnections. The results indicate that there is a possibility that at least some of the symptoms observed in AHS patients are the effects of network disconnections rather than direct grey matter lesions. More generally, the study proposes a new methodology regarding the approach to single cases that integrates a specific in-depth clinical evaluation, the analysis of previous clinical literature and an anatomical study which identifies the contribution of structures that are apparently uninjured but are disconnected from their networks and thus compromised as far as their functioning is concerned.

## 2. Methods

### 2.1. Case Report

BG is a 71-year-old, right-handed woman who had 13 years of education. She suffers from chronic arterial hypertension and diabetes mellitus. She had emergency cardiac surgery involving the dissection of the ascending aorta artery and the replacement of the aneurysm (Bentall’s surgery). Four days later, signs of psycho-motor agitation, sensorimotor deficits in the left upper limb and left visual field hemianopia appeared. A CT scan showed the presence of right cortical and subcortical hypodensity in temporo-parietal and occipital areas. Over the following weeks, her clinical condition improved and after 6 weeks from the lesion onset she was moved to a rehabilitation centre. At that time, she presented with left upper limb hyposthenia and severe somatosensory deficits in both the upper and lower limbs on the left. Furthermore, she showed visual field deficits (hemianopia) and spatial neglect. She was verbally fluent but sometimes appeared confused and showed signs of verbal perseverations. There were also some signs of AHS (see below for the assessment in the acute phase). The motor weakness in her left arm recovered in a few weeks, while the somatosensory deficits persisted.

When we met BG, she complained of experiencing difficulties with her left hand. She reported that the hand behaved in an uncontrolled manner and did not respond to her will; it interfered with or even tried to block the right hand when she was using that one and sometimes also disturbed other people who were nearby. Furthermore, her hand moved as if it was wandering in the surrounding space (i.e., levitation). These disorders impacted her daily life activities, for example, getting dressed, washing or doing housework, and were consistent with a diagnosis of Anarchic Hand Syndrome. These were checked at various intervals over time, and a comparison between BG’s symptoms and the symptoms reported in previous literature was made, taking into account the results of a previous review [2] and articles on the subject published from 2016 to 2020 (see Appendix A).

### 2.2. Anarchic Hand Syndrome Assessment

At the time BG was admitted to the rehabilitation centre, the weakness in her left arm persisted and the sensory deficits were still severe, in particular with regard to her left hand. Kinaesthetic sensitivity was only partially spared in her left arm (the shoulder, elbow and wrist) but not in her hand. In the scientific literature on AHS, the terms indicating the various symptoms associated with the syndrome are not always consistent, thus the patient’s symptoms were analysed according to a previously published classification of the symptoms that derives from a review of the literature [2]. This categorises 6 typologies of symptoms: (i) purposeful or semi-purposeful AH movements; (ii) non-purposeful AH movements; (iii) uncontrolled bilateral hand movements; (iv) hand-related feelings; (v) disconnection symptoms and (vi) other AH-related symptoms (see Table 1 for details). The clinical examination was repeated four times by the same examiners by means of clinical observations and interviews with the patient. In this way, the symptoms were checked for a period lasting up to 6 months after the lesion onset. In addition, in order to check for other cognitive disorders, a neuropsychological assessment was administered at 2 and 6 months after lesion onset (Table 2).

## 3. Neuroanatomical Study

### 3.1. Lesion Mapping

Five months after the stroke, the patient underwent a 3D-T1 and 3D-T2 radiological 3T MRI (Signa Architect, General Electrics, Milwaukee, WI, USA) examination. The lesion analysis was performed using the 3D-T1 MRI image. The lesion was manually drawn on the MRI axial slices and then reconstructed as a 3D region of interest (ROI) with MRIcroN [18]. The MRI scan and the ROI were then spatially normalised to a MNI template by means of SPM8 (Statistical Parametric Mapping; http://www.fil.ion.ucl.ac.uk/~spm, accessed on 6 March 2019).

To identify the grey matter structures encompassed by the lesion, BG’s normalised lesion was then compared with the AAL brain atlas [19] (Figure 1). The damaged white matter tracts were at first explored by means of Tractotron (part of the BCBToolkit; [20]; http://www.bcblab.com.html, accessed on 21 March 2019). The resulting drawing of the patient’s lesion was used as a ROI to track the white matter fibres based on a healthy subjects’ dataset [21], setting a threshold of 50% of overlapping maps for the localisation of the lesions [22,23]. Using this procedure, the severity of a tract disconnection can be established by assessing the proportion of the tract that is damaged. Grey and white matter structures damaged in at least 5% of their voxels were considered as candidates for further DTI analyses [24,25].

### 3.2. DTI Investigation

The second step of the neuroanatomical in vivo investigation of the white matter was conducted via a standard DTI tractography and a reconstruction of the tracts which had previously been identified as damaged by Tractotron.

A diffusion weighted imaging (DWI) sequence was carried out along with the 3D-T1 and 3D-T2 MRI imaging. This was done by means of the acquisition of 48 diffusion-weighted volume directions and 6 volumes with no diffusion gradient (B0) with a *b*-value of 2000 s/mm^2^. The images were obtained at a repetition time of 6000 ms and an echo time of 97 ms. The voxel size corresponded to 2 × 2 × 2 mm, the slice thickness was 2 mm, the FOV and the resolution were 256 mm and 128 × 128, respectively. One supplementary image with no diffusion gradient applied was collected, but with reversed phase-encode blips. This provided us with a pair of images with no diffusion gradient applied and with the distortions going in opposite directions. From these images, the susceptibility-induced off-resonance field was estimated and corrected using the TOPUP tool as implemented in FSL [26]. Subsequently, motion and geometrical distortion were corrected using the EDDY tool.

Hence, a computation of diffusion tensors and the whole-brain deterministic tractography were performed using StarTrack software (http://www.natbrainlab.com (accessed on 22 February 2019)). A damped Richardson-Lucy algorithm was applied for spherical deconvolutions. A fixed fibre response parameter of 1.5 was adopted, coupled with the geometric damping parameter of 8. Four hundred algorithm iterations were run and an absolute threshold of 0.0025 was adopted. A modified Euler algorithm was used to perform the whole-brain streamline tractography, with an angle threshold of 45°, a step size of 0.5 mm. Tract dissections were performed using Trackvis (http://www.trackvis.org/ (accessed on 16 November 2020)) in order to allow for an inter-hemispheric comparison of lesioned tracts and their spared homologues in the contra-lesional hemisphere [22].

Each tract that had previously been identified on Tractotron was reconstructed on the axial and coronal slices of the FA map by means of an approach related to the two ROIs (see Figure 2 for a detailed description of the tract reconstruction). The reconstruction of the corpus callosum (CC) was performed by dividing the midsection into seven parcels [27] corresponding to the rostrum, genu, anterior and middle body, posterior body, isthmus, and splenium of the CC. The seven-parcel division has recently been confirmed as being a good method of reflecting the functional division of the callosal connections [28].

## 4. Results

### 4.1. Anarchic Hand Syndrome

Table 1 shows the results of the repeated assessments of the evolution of AHS in the patient BG. In the acute phase, she presented with apparently purposeful movements, such as magnetic apraxia (i.e., the mere visual presence of an object near the hand—or touching the hand—triggers groping movements as well as grasping, with the patient being unable to inhibit this behavior) and grasping. Among the non-purposeful movements, there was evidence of levitation (i.e., an uncontrollable upward movement of the anarchic arm), while many symptoms indicating uncontrolled bilateral hand movements were present. In particular, there were symptoms suggesting: inter-manual conflict (i.e., the AH movements interfered with the actions of the non-AH, or the non-AH attempted to contrast the movements of the AH); diagnostic dyspraxia (i.e., uncontrolled cross-purposeful actions of the AH triggered by voluntary actions carried out by the non-AH); responsiveness to actions performed with the intact hand (i.e., AH movements are triggered by the non-AH starting an action or movement) and AH mirror movements (i.e., the AH reproduces the same movements as the non-AH). Other AH-related symptoms were unresponsiveness of the AH (i.e., the hand did not respond to the patient’s commands even though there was no evidence of paralysis); loss of dexterity and a deficit in bimanual coordination, and the tapping and performing of sequences. It was difficult to determine the presence of symptoms relating to disconnections. Indeed, BG presented with left hand apraxia, but her performance with her right hand was also at the inferior limit, and inter-manual sensory transmission was not testable due to her severe sensory deficits. Finally, autocriticism (i.e., the patient manifests exaggerated frustration with his/her body or hand), avoidance behavior (i.e., the patient unintentionally withdraws the affected hand from environmental contact or stimuli) and restraining actions (i.e., the patient tries to stop the AH, blocking it under his/her legs, putting it in a pocket or using the other hand to prevent it from moving) were also present.

At the following examination (90 days after lesion onset), the magnetic apraxia was resolved. Grasping, levitation and bilateral movements (inter-manual conflict, diagnostic dyspraxia and responsiveness) persisted, although these were less frequent than in the acute phase. Other symptoms such as responsiveness, dexterity deficits, bimanual incoordination and disorders in tapping and sequencing remained unmodified. The clinical conditions did not change in the two subsequent assessments.

A comparison with the previous literature on the topic (details are reported in the SM-B) shows that BG’s symptoms are typical of a number of different conditions. Indeed, some of them are significantly more frequent after posterior right hemisphere damage (i.e., levitation; sensory loss), while others are found in both anterior and posterior lesions (i.e., responsiveness, mirror movement, restraining actions, autocriticism, avoidance and deficits in dexterity). Finally, other symptoms are significantly more frequent in the presence of anterior lesions (i.e., magnetic apraxia, grasping, inter-manual conflict; unilateral apraxia and unresponsiveness).

### 4.2. Neuropsychological Assessment

At the time of the first neuropsychological assessment two months after the lesion onset, BG had recovered from her initial difficulties with spatial and temporal orientation and appeared to be aware of her deficits. In particular, she reported difficulties relating to concentration, mental flexibility and the control of automatic responses. As shown in Table 2, her general cognitive functions were not yet completely recovered (pathological scores in the ACE test: verbal fluency and visuo-spatial functions), although her scores in the MMSE [29] and coloured progressive matrices (CPM [30]) were above the cut-offs. Attention and executive functions were impaired, while verbal short- and long-term memory were preserved. BG’s mistakes in the Corsi spatial span [31] were probably at least in part influenced by spatial disorders. Indeed, the patient presented with signs of extra-personal neglect in both psychometric and behavioural tasks (BIT, [32]), while there was no evidence of personal neglect (Comb and Razor test, [33]). Furthermore, writing, reading and calculation were affected by spatial deficits, while language (relating to comprehension and expression) was unaffected. The extinction of the left-side stimulus, when presented along with a right-side stimulus (double stimulus condition [34]), was also present. Finally, BG showed left-hand ideo-motor apraxia (scores for the right hand were at the inferior limits for normal scores) and ideational apraxia.

At the check-up assessment six months after the lesion onset, BG’s general cognitive functions and neglect had recovered, as well as her difficulties relating to writing and reading. The deficits in attention and executive functions persisted. In addition, BG still presented with left apraxia.

**Table 2 brainsci-11-00632-t002:** Neurological and neuropsychological assessment. BG’s performance in sensory-motor and neuropsychological tasks carried out 2 and 6 months after lesion onset was reported. The scores are corrected for age, gender, and education.

Test	2 Months	6 Months
*Sensory-motor deficits*		
MRC Scale [35]	5-	5-
R-Nottingham Sensory Assessment [36] Left hand		
Light touch	**0**	**0**
Temperature	**1**	**1**
Pinprick	**0**	**1**
Pressure	**0**	**1**
Tactile localisation	**0**	**1**
Proprioception	**0**	**1**
Kynesthesia	**0**	**1**
Pain	**0**	**1**
*General cognitive functions*		
Addenbrooke’s Cognitive Examination (ACE-R)	**60.64**	76.11
Mini Mental State Examination (MMSE, [29])	24.86	28.86
Colored Progressive Matrice (CPM, [30])	19.4	27.1
*Attention and Executive functions*		
Attentional matrix [31]	n.a	**27.25**
Digit span forward [37]	5.13	**4.13**
Digit span backword [37]	**2.97**	3.08
Frontal Assessment Battery (FAB, [38])	**10.7**	**11.9**
*Memory*		
Pairs of words learning [39]	16.4	n.a
Story recall—short term [31]	6.7	n.a
Story recall—delayed [31]	6.5	n.a
Corsi Spatial span [31]	**3.25**	**3.25**
Corsi supraspan	n.a	11.34
*Spatial neglect*		
Behavioural Inattention Test (BIT, [32])		
Conventional	62	139
Behavioural	64	n.a
Comb and razor [33]	0-0	n.a
Visual extinction [34]		
Single stimulus on the right (max 10)	10	10
Ssingle stimulus on the left (max 10)	10	10
Double stimuli (20)	0	**0**
*Ideomotor apraxia* [40]		
Right limb	61	61
Left limb	**39**	**41**
Pantomiming the use of objects [40]	**14**	18

In bold are the pathological scores. n.a. = not administered tasks.

**Figure 2 brainsci-11-00632-f002:**
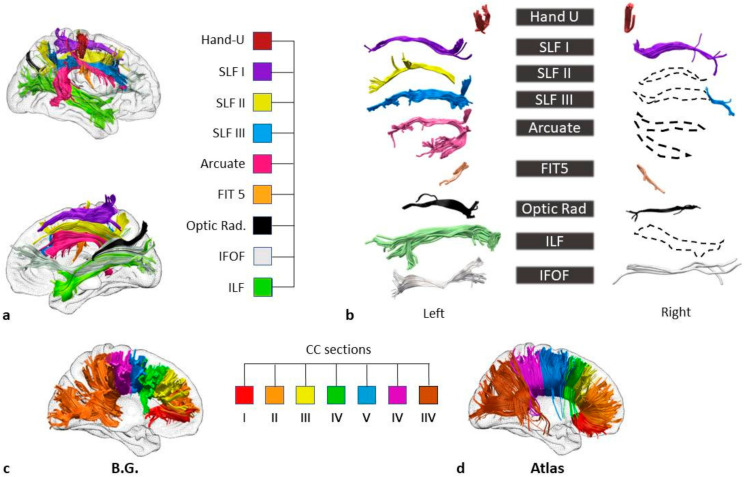
BG’s white matter disconnections. (**a**) Lateral (at the top) and medial views of the right hemisphere and the disconnected tracts (Tractotron analysis). The tracts were reconstructed based on a healthy subjects’ dataset [21] Hand-U= U-shaped tracts of the hand; SLF I = first branch (dorsal) of the superior longitudinal fasciculus; SLF II = second branch of the superior longitudinal fasciculus; SLF III = third (ventral) branch of the superior longitudinal fasciculus; FIT 5 = fronto-insular tract 5; Optic rad. = optic radiations; IFOF = inferior fronto-occipital fasciculus; ILF = inferior longitudinal fasciculus. (**b**) The reconstruction of tracts based on the patient’s diffusion weighted imaging. Dotted lines represent the tracts for which tracing was hindered by the lesion. In order to isolate the inferior fronto-occipital fasciculus (IFOF), the two ROIs were placed around the white matter of the occipital lobe and the anterior floor of the external/extreme capsule, respectively. For the reconstruction of the inferior longitudinal fasciculus (ILF), a first ROI was placed around the white matter of the frontal lobe, and the second ROI around the white matter of the occipital lobe. The reconstructions of the superior longitudinal fasciculus I (SLF I), II (SLF II), and III (SLF III) were carried out by drawing the first ROIs on the coronal slices of the superior, middle and precentral frontal gyri, respectively, and the second on the parietal lobe [41]. For the three segments of the arcuate fasciculus, the ROIs were placed on: (i) the precentral gyrus and postcentral gyrus for the anterior segment; (ii) the pars opercularis and the auditory cortex for the long segment and (iii) the inferior parietal lobe and the auditory cortex for the posterior segment [42]. The fronto-insular tract 5 (FIT5) was reconstructed via two ROIs on the subcentral gyrus and the anterior long gyrus of the insula [21] ROIs on the parietal postcentral gyrus and the frontal precentral gyrus allowed for a reconstruction of the Hand U-shaped fibres. The reconstruction of the optic radiations was carried out with the ROIs placed on the lateral geniculate nucleus and the occipital cortex. (**c**) The reconstruction of BG’s corpus callosum in seven parcels equally divide the midsection, according to Witelson’s [27] division. (**d**) The reconstruction of the seven parcels of the corpus callosum performed on a healthy subjects’ dataset [21].

## 5. Neuroanatomical Results

The comparison between AP’s lesions and the AAL template is shown in Figure 1.

The patient’s lesion in the right hemisphere extends out from pericentral regions, the insula and the temporal lobe to the inferior parietal and occipital cortices. No direct damage to the frontal cortex was recorded.

However, a first analysis of the white matter (Tractotron) showed that the white matter damage had expanded beyond the areas of direct lesion, involving long fronto-parietal, fronto-occipital and temporo-occipital connections, the corpus callosum, and short sensorimotor connections. The analyses of the DTI results (Figure 2a,b) confirm this and made it possible to identify the specific tracts which were disconnected in BG. A reconstruction of the three segments of the arcuate fasciculus on the right hemisphere was prevented by the extent of the lesion, confirming a disconnection in this tract. This means that there were disconnections between: (i) the ventral portion of the precentral gyrus and the postcentral gyrus (Anterior Segment), (ii) the pars opercularis and the auditory cortex in the superior temporal gyrus (Long Segment) and (iii) the Posterior Segment that links the inferior parietal lobule to the superior temporal gyrus.

The extent of the lesion also compromised the reconstruction of the SLF in the right hemisphere. The damage involved the three branches, with disconnections between the superior parietal lobe and the superior frontal gyrus (SLFI), the angular gyrus and the posterior regions of the middle frontal gyrus (SLF II), and the supramarginal gyrus and the inferior frontal gyrus (SLF III). Among the long tracts, the IFOF was compromised by the lesion (Figure 2b) with a disconnection between the orbital and the occipital cortices, as well as the ILF, with a disconnection between the anterior temporal lobe and the occipital cortex (Figure 2b).

Some short tracts were also damaged, in particular the FIT5 which connects the insula with the subcentral gyrus, the optic radiations (from the lateral geniculate nucleus of the thalamus to the occipital cortex) and Hand U-shaped tracts, with disconnections among the sensory and motor regions of the hand.

Finally, the reconstruction of the corpus callosum (CC) into seven parcels revealed damage to the posterior body of the structure (see Figure 2c,d) corresponding to the interhemispheric link between the bilateral somatosensory and motor regions [43,44].

Taken as a whole, these neuroanatomical results indicate that a lesion that directly involves only the post-rolandic posterior areas may indeed hide a much wider impairment in brain networks as a result of the widespread disconnections between structures which are located farther away.

## 6. Discussion

The results from our research into this single case derive from an integration of three different sources of information: (i) the clinical assessment, including a neuropsychological evaluation; (ii) a neuroanatomical analysis capable of integrating data from the direct lesions and white matter disconnections and (iii) a revision of the evidence already present in the literature on the topic. We consider that this approach has led to two main results: (i) evidence of the benefits of this type of methodology which overcomes the limits inherent in using each of these individual standpoints in isolation and (ii) a better understanding of the neural mechanisms underlying the various different symptoms of AHS indicating that there are not necessarily multiple subtypes of the syndrome as previously believed (see also below).

The clinical investigation of an individual patient’s deficits represents the main source of knowledge in the field of neuropsychology and remains the best approach in order to achieve an accurate, thorough study of rare syndromes such as this. In fact, single-case studies allow the researcher to analyse a patient’s symptoms in such depth and with an accuracy that group studies do not really provide since tasks and procedures need to be standardised. As a consequence, establishing the correlation between symptoms and lesion sites constitutes the most important type of procedure in neuropsychology. However, recent advances relating to the techniques used for the collection and analysis of neuroimaging data indicate that if neuroanatomical investigations are limited to the mapping of lesions (i.e., an analysis of the direct, discrete lesions), there is a risk that the work on the brain is oversimplified and this may impede the understanding of those symptoms that are due to functional impairments in regions that are distant from the site of the damage [15,17]. Diaschisis (i.e., a dysfunction in a distant brain region connected to the damaged areas, [45,46]) and disconnection (i.e., a dysfunction in two intact areas connected by a damaged tract [42,47]) may destroy networks that are necessary for specific functions. Although this appears to be an important step forward in the field of neuropsychology, a further step is necessary when rare syndromes are being analysed, namely a comparison with other related cases present in the literature. This is important in terms of replicability of data as, due to extreme inter-individual variability, the results from a single case may be due to one single specificity in the individual’s neuroanatomy rather than the result of the organisation of the general networks. Taking this factor into account, we therefore did not limit our study to the single case and a study of the neuroanatomy relating to the individual but also compared our results with previous related cases reported in the literature and ascertained the statistical frequency of each type of symptom in the various lesion sites for comparison purposes. This allowed us to identify the symptoms which might be considered as being typical of anterior, posterior, mixed or callosal forms of AHS and determine how to categorise them.

### 6.1. Does a Study of the Various Symptoms Associated with the Anarchic Hand Syndrome Make It Possible to Distinguish Different Subtypes of the Syndrome?

AHS is considered to be a multifaceted syndrome or even a group of partially separated syndromes, each characterised by a specific cluster of signs and putative neuroanatomical correlates [2,5]. The existence of callosal and frontal subtypes has been suggested ([2,5] for a revision of the literature). In addition, although less frequent, AHS has been described after damage to the posterior cortical regions [2,13,48,49,50] or subcortical lesions [11].

According to the results of the present study, we may conclude that any classification of AHS subtypes may be the effect of partial neuroanatomical investigations and that any distinction identified, at least for the “anterior” and “posterior” forms, may be fallacious. Indeed, only certain symptoms are significantly more frequent in one form than any other when the frequencies are statistically compared (see SM-B).

The analysis that was carried out of BG’s lesion map indicated that her deficit could be considered to be a posterior form of AHS. As confirmation of this, there are indeed certain symptoms that are typically associated with the posterior subtype of the syndrome, such as sensory loss and levitation. In contrast, she did not report feelings of alienness or personification. She manifested evidence of restraining actions, autocriticism and avoidance behaviour, which according to our analysis are more frequent in relation to the posterior than callosal lesions but not to the anterior lesions.

BG also displayed other symptoms, such as magnetic apraxia and grasping, although these might also be typical of anterior, frontal lesions. In addition, BG presented with uncontrolled bilateral hand movements [2]. Among these, as in the previous cases which we analysed, inter-manual conflict is more frequent in relation to anterior than posterior lesions and in posterior rather than callosal lesions. Diagnostic apraxia and responsiveness to actions performed with the intact hand are reported less frequently than inter-manual conflict, with diagnostic dyspraxia being more often linked to callosal lesions. In BG, these symptoms persisted over time. Lack of bimanual coordination, loss of manual dexterity and rhythm disorders are usually associated with damage to the corpus callosum [5] and in general with anterior fronto-callosal lesions.

Finally, there was no definitive evidence that unilateral apraxia was one of BG’s symptoms as, although her performance was worse for her left than her right hand, her scores for the right hand were just above the cut-off, indicating a potential index of some bilateral deficits. However, it is worth noting that BG is not left-handed and her right hemisphere symptoms (e.g., neglect and visual memory deficits) are not consistent with function-crossed laterality. The literature indicates that when unilateral apraxia is the consequence of a fronto-callosal lesion, it may involve both the right or left hand, while unilateral left apraxia is associated with posterior and callosal lesions [2]. Thus, this symptom in BG is consistent with both a localised posterior lesion and more extensive damage involving a disconnection between the antero-posterior tracts.

Finally, BG’s cognitive deficits are mixed. In effect, she suffers from visual neglect and disorders in visual attention and memory which are consistent with posterior right hemisphere damage, but evidence of a dysexecutive syndrome is also present, which is typical of frontal damage. As the impact of non-motor factors on the mechanisms relating to motor control have been documented [2,10], we cannot exclude the possibility that these cognitive disorders have had an influence on the AHS symptoms.

Taken as a whole, these data would indicate a mixed form of AHS, with symptoms typically associated with the anterior, posterior and callosal forms. However, this is inconsistent with the neuroanatomical data coming from the lesion mapping that was carried out which show a lesion in the posterior part of the right hemisphere. An in-depth analysis of the networks disconnections allowed us to disambiguate this contrast.

### 6.2. How the Analysis of the Disconnections Helps Us to Understand AHS Symptoms Better

The results from the lesion mapping indicate the involvement of posterior regions in BG’s lesion. Cases of AHS after lesions affecting the parietal cortex have previously been documented ([2] and SM for a review) with symptoms, such as non-purposeful movements, that were also present in our patient. In fact, the parietal cortex is associated with the awareness of the intention to move [10,51], also described as the “willing to move” signal [52], and the detection of motor errors [53,54]. Levitation (Table 2, Appendix A) may also be in some way an expression of the presence of a disorder in the awareness of the intention to move, as the anarchic hand moves around without and even in contrast with the subject’s will. However, the posterior damage only partially accounts for the variety of clinical manifestations displayed by BG. As previously discussed, along with typical posterior symptoms, this patient shows mixed symptoms ascribable to more anterior damage. The combination of the classic lesion mapping with the white matter disconnection mapping allowed us to identify all of the anatomical networks involved in BG’s lesion, such as the fronto-parietal attentional network and the networks underlying the sense of agency, motor control and action planning. These findings may account for the complexity of the symptoms observed in BG and confirm previous evidence indicating that the motor awareness system is associated with the integration of multiple large networks [10,16].

In BG, the fronto-parietal disconnection is due to damage to the three branches of the SLF and the anterior segment of the arcuate fasciculus. In the absence of an anterior lesion, these disconnections explain BG’s symptoms relating to environmental dependence and lack of inhibitory control (e.g., grasping, diagnostic dyspraxia and responsiveness). The same disconnections also explain BG’s cognitive profile, in particular her dysexecutive symptoms and neglect. The latter is probably also due to damage to the optic radiations and ILF [23,55].

Previous findings in an AHS patient [12] revealed that parietal damage may lead to an abnormal sense of agency, which is probably associated with the pivotal role of this area in the multisensory integration of spatial, temporal and sensory contingencies [56]. AHS patients not only deny being the agent of the anarchic limb movements (i.e., with statements such as “My hand moves according to its own will”), but at times also display a reduced sense of agency for their voluntary movements [12]. However, a recent study [57] showed that during self-other attribution of agency, the parietal cortex is jointly activated with the primary motor cortex and the middle temporal gyrus. Thus, rather than a single lesion, it could well be the antero-posterior disconnection between these structures which plays a role in the external attribution of agency of the anarchic hand movements. The sense of agency is also modulated by the activity of the insula, which has been extensively described as a hub for the processing of internal bodily and emotional states and is disconnected in BG (via FIT5).

Finally, the posterior damage to the corpus callosum as shown in BG’s tractography is in line with the previous literature that describes the emergence of inter-manual conflict and diagnostic dyspraxia in AHS patients after callosal lesions. Moreover, the unilateral apraxia displayed by BG may be linked to the callosal disconnection, as cases of unilateral apraxia after damage to the posterior part of the corpus callosum have previously been documented [58].

This study is subject to the limitations which are intrinsic to all single case studies. The collection of data from a cohort of patients displaying different variants of AHS would provide stronger evidence of the clinical and anatomical profile relating to the syndrome. Nevertheless, it is important to take into consideration that AHS patients are extremely rare, and to the best of our knowledge there are very few reports that integrate neuropsychological and neuroanatomical investigations and include a comparison with previous literature on the topic [2]. We thus hope that there will be further studies which will take advantage of the methodologies used in this work in order to expand and improve on existing knowledge regarding this syndrome. Another limit is the lack of neuroanatomical data from healthy controls. As a result, the patient’s lesions were compared to data from an anatomical atlas (AAL, http://www.cyceron.fr/web/aalanatomicalautomaticlabeling.html (accessed on 11 March 2019)) and the tractography reconstruction of one hemisphere was compared with the reconstruction of its homologue (a procedure already followed in previous studies; [17,23]). Finally, we also consider that the patient’s MRI shows signs of cortical atrophy that have not yet been specifically investigated. However, the repeated clinical and neuropsychological assessments did not reveal symptoms of dementia and the cognitive profile remained stable over time.

## 7. Conclusions

This study suggests that the classic distinction of AHS symptoms according to the location of the lesion, that is, anterior or posterior, does not fit in with the diversity of clinical manifestations displayed by the patients. It also underestimates the effects of structures that, although not directly damaged, are dysfunctional due to a disconnection in their networks. The study takes advantage of (i) an AHS symptom classification which resulted from an up-to-date review of the literature; (ii) a repeated investigation of the patient’s symptoms (developed based on the previous literature on the topic) and (iii) an in-depth anatomical investigation conducted via the integration of the study of direct lesions and white matter disconnections. Based on our results, we consider that AHS should be considered as a disconnection syndrome, whose symptoms are associated not only to the effects of direct lesions in the grey matter but also to the disconnections of white matter long and short tracts.

The classification of AHS symptoms into sub-categories is useful for the purpose of identifying each patient’s clinical picture via a comparison with previous data regarding the frequency of symptoms associated with different lesion locations. However, a specific assessment would test all the symptoms, not only those that are expected as a result of the lesion site.

From a methodological point of view, single case studies benefit from the integration of data from neuropsychological assessments, comparisons with previous literature on the topic and an in-depth neuroanatomical analysis which consider and integrate results from direct grey matter lesions and white matter disconnections.

## Figures and Tables

**Figure 1 brainsci-11-00632-f001:**
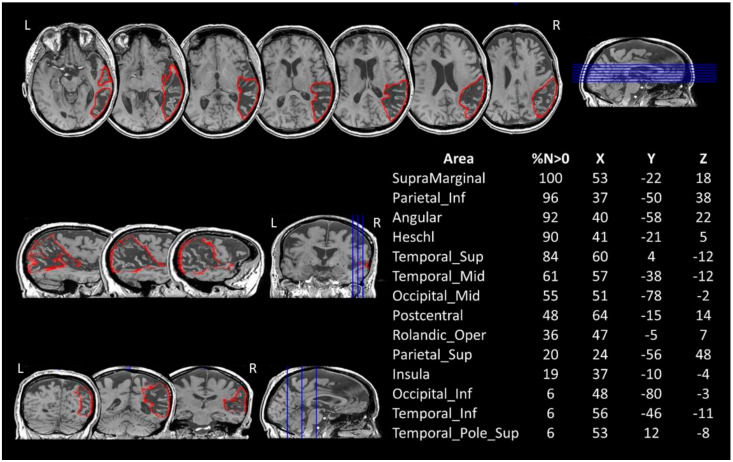
BG’s 3D-T1 MRI image and lesion mapping. Axial (**up**), sagittal (**middle-left**) and coronal (**bottom-left**) views are shown with the drawing of the lesion in the right hemisphere in red. The table on the right reports the percentage volume (%*N* > 0) affected by the lesion and the MNI coordinates (X, Y, Z) for each grey matter structure, as reported on MRIcron (AAL atlas). Lesion’s centre of mass = 140.42 × 84.64 × 97.34. L = left. R = right.

**Table 1 brainsci-11-00632-t001:** BG’s AHS symptoms as assessed in the repeated sections of evaluation.

Symptom Category Symptoms		50 Days	90 Days	154 Days	180 Days
**Purposeful AH movements**	*Magnetic apraxia*	+	−	−	−
	*Grasping*	+	+/−	+/−	+/−
	Forced Grasp	−	−	−	−
	Groping	−	−	−	−
	Other	−	−	−	−
**Non−purposeful AH movements**	Exploratory behaviour	−	−	−	−
	Repetitive movements	−	−	−	−
	Self grabbing	−	−	−	−
	Levitation	+	+/−	+/−	+/−
	Nocturnal movements	−	−	−	−
**Uncontrolled bilateral hand movements**	*Intermanual conflict*	+	+/−	+/−	+/−
	*Diagonstic Dyspraxia*	+	+/−	+/−	+/−
	Responsiveness	+	+/−	+/−	+/−
	Mirror, Synkinesias	Mir	−	−	−
**Hand-related Feelings**	Alienness, Personification	−	−	−	−
	Rest, Autocriticism, Avoidance	A/R/V	R	R	R
**Disconnection symptoms**	*Unilateral apraxia, agraphia*	Ap+	Ap+	Ap+	Ap+
	Inter-manual sensory trasmission, Tactile agnosia	imp	imp	imp	imp
**Other AH-related symptoms **	*Unresponsiveness*	+	+	+	+
	Dexterity	+	+	+	+
	*Bim Inc/Tapping/Sequence*	Bi Inc/Tap/Seq	Bi Inc/Tap/Seq	Bi Inc/Tap/Seq	Bi Inc/Tap/Seq
	Sensory loss	+	+	+	+
	Lower limb hyp	−	−	−	−
	Mutism (initial)	−	−	−	−
**Cognitive deficits**	Language dis	na	−	−	−
	Verbal memory	na	−	−	−
	Visual memory	na	+	na	+
	Writing, reading, calculation	na	+	na	−
	Attentional dis	na	+	na	+
	Neglect	na	+	na	−
	*Disexecutive syndrome*	na	+	+	+
	Ideational Limb Apraxia	na	+/−	na	+

In italics are the symptoms usually associated with anterior lesions. + = presence of the symptom; − = absence of the symptom; +/− = symptom not stable (this is reported to happen only in certain moments); na = not assessed; Imp= impossible to assess; Mir = mirror movements; A = autocriticism; V = avoidance; R = restraining actions; Ap = apraxia; Bi Inc = bimanual incoordination; Tap = failure in tapping; Seq = errors in action sequences; hyp = hyposthenia; dis = disorders.

## Data Availability

The data are all shown in the paper.

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
