# Peer review of "The Role of White Matter Disconnection in the Symptoms Relating to the Anarchic Hand Syndrome: A Single Case Study"

_brainsci, 2021, doi:10.3390/brainsci11050632_

Round 1

Reviewer 1 Report

This article presents a case report of a subject diagnosed with anarchic hand syndrome. The topic is of interest, as this syndrome is not commonly found in the clinical setting. In addition, when it manifests in a patient, it is difficult to clarify the specific neurological substrates. In this regard the authors propose an innovative methodology to understand and detect the specific brain areas and white matter tracts that are affected. This information is crucial for further rehabilitation therapies as it allows not only to better understand the problem but to stablish personalized therapies. This approach could be of value when dealing in the clinical environment with single subjects with no means to compare the subject findings with a normative population.

            The case report is presented in great detail and explains in a step-wise manner the process that was carried out by the researchers and clinicians for the patient assessment and to reach their conclusions.

            From my point of view only minor changes and clarifications are necessary. Those are detailed in the next paragraphs.

  • Methods Section:
  • The authors state that in the patient “left visual field deficits appeared”. It would be of interest to explain them further (i.e. blurred vision, hemianopia…).

  • Neuroanatomical study: were the acquired T1-wi and T2-wi images 3D sequences with high spatial resolution? Or were they a 2D for clinical assessment purposes? I guess, according to the posterior analysis, that at least the T1-wi was a 3D-T1-wi. Please specify.

  • Diffusion Tensor imaging: the authors used a b factor of 2000 s/mm2, instead of 1000 s/mm2 which is more commonly reported in literature. I am sure there is a specific reason for the chosen b value that I would like to know. DTI is a technique based on the application of strong gradient fields which cause an attenuation of the measured MR signal. It has been shown that an increase in the b value generally leads to the decrease of signal-to-noise-ratio (SNR) due to the application of larger diffusion gradients. The reduction of SNR has a detrimental effect on diffusion parameters causing an upward bias in FA particularly if the true FA is low, as for example in grey matter or in this case in the lesion area and surrounding tissue. It also reduces the certainty in the principal direction of diffusion. In this sense, how can the authors be sure that part of the observed damage in the WM tracts is not related to the b value?

  • Results:
  • Table 1: please provide in the Table legend the full meaning of abbreviations such as language dis, verbal mem, vis mem…etc.
  • Figure 1: please provide in the image some labels to indicate the Left and Right hemispheres.
  • Figure 1, Table: the Table provides “the percentage volume affected by the lesion and their coordinates”. I am not sure how this information was obtained (software, procedure), can you explain it further?
  • The authors state that right arcuate fasciculus could not be reconstructed by the extent of the lesion, what about the left hemisphere? Did you find the long segment there? It could be of interest to explain it in the text as the patient presented some language deficits.
  • Figure 2: it seems that in Figure b there IFOF and ILF labels are exchanged.
  • Related to the reconstruction of the CC callosum, what happened to the temporal connections. Were they also affected? I am not able to see them properly on the segmentations, and according to the lesion location it is very likely that they were affected at least in the right hemisphere. Please clarify.

  • Conclusions:
  • The authors state: “This study demonstrates that the classic distinction of AHS symptoms according to the location of the lesion, that is, anterior or posterior…”. This affirmation is a little bold taking into account that the results are drawn from a single subject. I think it would be more appropriate to change this affirmation to “this study suggest…”.

Reviewer 2 Report

The authors carry out research on a single case of anarchic hand syndrome and integrate multiple methods in the assessment of the patient's conditions. Their results lead to conclude on the advantage of combining multiple methods, overcoming limitations inherent to individual standpoints used in isolation. Thus, the work offers an improved understanding on the mechanisms of pathophysiology underlying the patient's symptoms. 

Results are displayed clearly and explained exhaustively. Conclusions are sufficiently supported by the discussion. I enjoyed reading this work and I found the combined approach insightful and useful to inspire future studies. 

Minor remarks:

In the result section: please clarify whether, 90 days after lesion onset, "magnetic apraxia was resolved" or whether "these symptoms persisted": the current phrasing is slightly ambiguous and it might be improved by  specific statements about which symptoms were resolved and which ones endured (as nicely shown in the table).

In discussion, "How the analysis of the disconnections helped us to understand AHS symptoms better": please revise the text in the fifth line: "The detection of motor error." the words do not constitute a self-standing sentence. 

In the same paragraph, authors state that "in absence of anterior lesion, these disconnections explain BG's symptoms ... (e.g. magnetic apraxia...". Please revise this passage: I struggle to follow your inference as your imaging data are acquired 5 months after stroke, but the symptom magnetic apraxia is resolved by that time. If disconnection explains magnetic apraxia as symptom, should not the symptom be present 5 months after stroke?

Notwithstanding the advantage of integrating multiple sources of information, the accurate and interesting description of symptoms over time is complemented by lesion mapping data 5 months after stroke. In my opinion, comparing neuropsychological evaluation and imaging data over multiple times after injury could add further insight and, ideally, it may be considered in future studies.
